# Reconsidering Homosexual Unification in Islam: A Revisionist Analysis of Post-Colonialism, Constructivism and Essentialism

**Aisya Aymanee M. Zaharin**

School of Social Sciences, University of Queensland, Brisbane, QLD 4072, Australia;
a.zaharin@uq.edu.au or aisya.zaharin@uqconnect.edu.au

**Abstract:** While there are Muslims who hold that the two identities Muslim and homosexual are mutually exclusive and that it is illegal to practise homosexual acts, the ongoing Muslim revisionist movement seems to provide a more understanding approach. Members of this movement argue that although the act of homosexuality is mentioned in the *Qur'ān*, this term is not inherently similar to the contemporary understanding of homosexuality based on love (mawaddah) and mutual consent. This paper provides a comprehensive understanding of theological, historical and sociological discourses to demonstrate some of the challenges facing contemporary sexual ethics in relation to Muslim homosexuality and its relationship to power (religious, patriarchal and neo-colonialist). It argues that conservative and centrist scholars have always presented homosexuality based on cis-hetero morality standards, but not from a deep understanding of the *Qur'ān* and the *Hadith*. It also argues that from a conservative view, homosexuality has never been addressed from either a medical/psychological (essentialism) or a social constructivist perspective, thereby further denying the existence of men who are not attracted to females as explained in the *Qur'ān* (24:31). On a historical level, this paper offers a discussion of legal, social and political genealogies of the history of same-sex attraction among Muslims by addressing relevant questions related to pre-colonial and post-colonial legacies. This paper then considers sociological and scientific approaches (constructivism and essentialism theories) that explain a place for same-sex unions in Islam.

**Keywords:** homosexuality; Islam; revisionist Muslim; decolonisation; LGBTQIA+; post-colonialism; Islamic studies



## 1. Introduction

The discussion of homosexuality in Islam has been subject to widespread anathema among Muslim communities in the present day. There are both Muslims and non-Muslims who believe being a homosexual and a Muslim are mutually exclusive identities (Hammoud-Beckett 2022; Pallotta-Chiarolli et al. 2021; Zaharin and Pallotta-Chiarolli 2020). Most medieval Muslim scholars, except for *Hanafis*, unanimously agreed that the punishment for homosexuality in Islam should be exactly the same as the *hadd*[1] for heterosexual adultery. For the *Hanafi* School, the fact that male same-sex acts are not mentioned as punishable acts in the *Qur'ān* excluded them from the *hadd* category. The *Hanafis* argued that sodomy should be punished at the judge's discretion (*ta'azir*) rather than the fixed *hadd* being applied, as in adultery (Peters 2005, p. 61). Islamic law is based on interpretations of *Shari'a*, which is an interpretation of the *Qur'ān* and the *Hadith*. Most contemporary Muslims today follow the interpretation of the *Qur'ān* found in certain Sunni and Shia sects that believe homosexuality is the most punishable sin and deserves the death sentence as well as calamity from God. Regardless of the varying sentences applied, homosexuality in the present day is formally not condoned in any Muslim country (Zaharin and Pallotta-Chiarolli 2020). Over the past two decades, however, revisionist Muslim scholars, such as Jamal (2001), El Fadl (2002, 2014), Habib (2008, 2010), Kugle (2010a), Hamzic (2015), Naraghi (2015), Jahangir and Abdullatif (2016, 2018) and Siraj (2016), have presented a more tolerant

approach by discussing the homosexuality issue in a more detailed and nuanced way. They explain that the act of Lut's people in the *Qur'ān* is different from the current understanding of homosexuality. The relationship between Islam and homoeroticism cannot be simplified as a one-dimensional issue of "Islam versus the LGBTQI+". Henceforth, this paper invites the reader to analyse previous Muslim society's life and values through the analysis of queer and homoerotic historical evidence produced in the Muslim world between the 11th and 21st centuries as one of the ways to understand the lives of queer Muslims in the past and the present.

This paper further aims to highlight the progressive and revisionist Muslim standpoint by outlining several reasons why homosexuality should be tolerated by Muslims today. In doing so, it challenges the current Muslim understanding of homosexuality and its relationship to power (religious, patriarchal and neo-colonialist). It presents the conservative, the centrist and the reformist perspectives by looking into theological debates on the permissibility and relevance of same-sex sexual desire and conduct. It confronts the conservative view of homosexuality as abnormal, deviant and immoral behaviour and supports the revisionist Muslim view. This is achieved by addressing the historical, social and political genealogies of the Islamic legal tradition from *Hadith* and *Qur'ān* theoretical perspectives and the history of same-sex attraction among Muslims in the past and present. It argues that conservative and centrist scholars have always presented homosexuality based on cis-hetero morality standards, but not from a deep understanding of the *Qur'ān* and the *Hadith*. It also argues that homosexuality has never been addressed from either medical and psychological (essentialism) or social constructivist reasoning, thereby further denying the fact that there are men who are not attracted to women as explained in the *Qur'ān* (24:31). Next, it offers a critical pre- and post-colonial historical analysis of the discourses on homosexuality related to contemporary Islamic law. It explains how the colonial matrix of power is still continuously affecting the colonised nations and, therefore, has become the major point of animosity towards the LGBTQ+ community in Islam in the present day. By connecting the current Islamic understanding of homosexuality and the effect of post-colonialism, this paper further explains how this effect perpetuates human rights abuses upon Muslim LGBTQ+ today. The paper then augments these discourses with a sociological and scientific approach by analysing constructivism and essentialism to support a possible place for same-sex unions in Islam as described by Jahangir and Abdullatif (2016, 2018). Finally, this paper is not interested in redefining homosexuality as a sin but rather it is revisiting the *Qur'ān* verses and other revisionist scholars' clarification of the homosexuality issue. At the same time, this paper also emphasises that being gay and Muslim are not mutually exclusive—for they have existed since time immemorial and same-sex attraction is naturally inherent.

## 2. Homosexuality from the Muslim Revisionist View

I have written this paper for Muslims who understand the spirit of Islam, which is to generate unified societies and are willing to challenge their own understandings in order to promote religious and personal growth. It is for those who understand that the very foundation of Islam is about acquiring knowledge to read and rethink (96:1), understand God's vastness and the greatness of Allah's powerful knowledge (42:50) and understand Allah's capability that is beyond human grasp. This paper might not sit well with Muslims whose definition of right and wrong is purely black and white (*Halal* and *Haram*), Muslims indiscriminately parroting conservatism and defining Islam exclusively for cis-hetero and "holier than thou" Muslims. This paper stands with the revisionist Muslim position as described in Ali (2016), Kugle (2010a) and Alipour (2017) that homosexuality, as a biological variation, or even as an early childhood experience, is not an individual's rational choice and therefore should be accepted as God's divine will as part of the variety of humankind mentioned in the *Qur'ān*. Hence, this paper does not focus on a discussion of whether homosexual orientation is a test from Allah or a "divine trial" (Al-Husaini 2005) and if those who pass this massive test will be rewarded immensely in the hereafter, as some

Muslims have wondered. Permanent celibacy is not part of Islamic teaching, and the notion of such a "test" seems inconsistent with "divine justice" (Dabashi 2008; Esack 1997), as in why the kind-hearted Allah wants homosexual people who are naturally born with such an instinct to be punished with needless suffering. In addition, since test or tribulation in Islam is often associated with people who have strong faith, does this mean that LGBTQ+ people are imbued with stronger faith?

This paper then concludes that both arguments (1) that homosexuality is a social construct (constructivism) and (2) that homosexuality is an inherent (essentialism) sexual behaviour support the revisionist view on homosexuality in Islam. By presenting both arguments, this paper attempts to close the gap between the essentialist and the constructivist argument among revisionist scholars in order to provide a more nuanced understanding of homosexuality in Islam.

While not necessarily the same homosexual discourse as in the *Qur'ān*, Islam has a long, recorded history of normalising homoeroticism and sexual behaviour between members of the same sex (male especially) that can be seen in poetry, literature and paintings in the major languages of the Muslim World (Duran 1993; Rowson 2008; Bouhdiba 2004; Kligerman 2007). There is also no record of the prophet having anyone punished for homosexuality (Shinqiti 2008; Hendricks 2010). Interestingly, while present-day Muslims assume that homosexual behaviour was outlawed by Islam, it was not until after the colonial period of the late 18th and 19th centuries that this conservative opinion became prominent. Scholars such as Dunne (1998), Kligerman (2007), Kugle (2010a), Lapidus and Lena (2014) and Nawaz (2016) insist that intolerance towards queerness and homosexuality was introduced to the Muslim world during the period of European colonialism. Nevertheless, Muslims in the past did not officially accept or frame homosexual behaviour as it is in the present day. Homosexual behaviour was socially common among elites[2] during the Mughal, Abbasid, Mamluk, Safavids and Ottoman empires but still took place within the domain of heteronormativity, along with the prevalence of slave ownership, during the medieval and pre-colonial Islam. It mainly existed on the condition that the elites would continue to publicly live a heterosexual lifestyle by having several wives and children. It still followed the traditional gender and power patterns where boys (or the slaves) played the passive role (matching women), while adult males asserted their power by receiving sexual pleasure through domination (Kligerman 2007). As such, upper-class men in Muslim empires often had young males who often played the passive role in the relationship. During medieval Islam, the openness to sexual relationships with one's slaves was one of the major loopholes for the occurrence of homosexuality, since the master was "permitted" to make equal use of his young male slaves as he did of the female slaves (Boronha 2014). By decree, a master was not permitted to have sex with male slaves. He simply flouted rules with impunity since he was allowed to do so with female slaves. For instance, Sultan Al-Fateh, the Ottoman conqueror of Constantinople, and Sultan Mahmud Ghaznawi of the Mughal Empire are both known to have expressed homoeroticism with their young subordinates (Hamzic 2015; Kligerman 2007). Subsequently, while there is evidence that homoeroticism occurred, it may or may not fit (depends on the context) into the present-day constructions of homosexual relationships. Nonetheless, this homoeroticism narrative is often hidden from the Muslim story today as part of efforts to maintain conservative dominance consistent with the current *Shari'a*, as well as to erase the LGBTQ+ identity from Islamic history (comprised of *mukhannathun* (effeminate men), *khuntha mushkil* (indeterminate hermaphrodite), *mutarajjilat* (mannish women) and *Baghghā* (passive homosexual), among others) by virtuously blaming the Western culture. Thus, while traditional Islam does not have much direct relevance for contemporary same-sex relationships due to the heteronormativity of the earlier society, the question remains, should a Muslim in the 21st century accept both homosexual orientation and practice in order to provide a place for Muslims with homosexual attractions?

### 3. *Qur'ānic* Interpretation of the Story of Prophet Lut

The *Qur'ān* refers 14 times to the story of Prophet Lut, and it is used in both conservative and reformist and progressive Islamic discourses to understand or to criticise same-sex relationships. Mainstream traditionalist Muslim scholars conclude that Lut's people were punished purely because they engaged in sodomy between men. For them, the word *fahisha* (indecency/lewdness) in the story of Lut (*Qur'ān*, 29:28–30) is interpreted as the homosexual actions committed by the men of Lut. Interestingly, while the term *liwāṭ* (sodomy) finds no mention in the *Qur'ān*, the Arabic word *lūṭī* لوطي in the *Qur'ān* (for example, Ash-Shu'ara 26:160) refers to the people of Lut (c *ahl Lut*) and, therefore, the Arabic use of the derived noun *liwāṭ* لواط from the story of the people of Lut in reference to sodomy. However, revisionist Muslim scholars argue that the story of Lut in the *Qur'ān* does not represent the contemporary understanding of homosexuality based on love (*mawaddah*) and mutual understanding. The *fahisha* (lewdness) in reference to sodomy mentioned in the *Qur'ān* was condemned as an act of sexual violence committed purely based on *shahwa* (desire) to show domination by anally penetrating other men (Jahangir and Abdullatif 2018). Moreover, Kugle (2010b) states that there is no correlation between adultery and *liwāṭ* mentioned in the *Qur'ān* nor a specific verse that declares anal sex as being a *hadd* crime as explained by *Hanafi*'s jurists. Based on a revisionist Muslim standpoint on the *Qur'ān* and the *Hadith*, this paper presents several explanations for why Muslims in the 21st century should consider accepting homosexuality in Islam.

*3.1. Homosexuality as a Sexual Orientation or Identity Has Never Been Explicitly Mentioned in the Qur'ān*

When dealing with the homosexuality issue in Islam, a Muslim could possibly follow either (1) the majority traditional stance described by Al-Qaradawi (2001) that rejects all kinds of non-hetero identities, feelings and practices; (2) a centrist moderate stance such as described by Ramadan (2009) and Khan (2013) that accepts the feelings and identity of homoeroticism, but rejects the homosexual act (penetration/relationship) itself; or (3) the progressive/revisionist Islamic view that accepts homosexual identity and practice on the basis of interpretations of the *Qur'ān* and theological reflection (Kugle 2010a; Siraj 2016; Ali 2016; Alipour 2017; Jahangir and Abdullatif 2016, 2018).

As early as the Islamic classical period, the Arabic term *lūṭī* in the *Qur'ān* Ash-Shu'ara (26:160) was defined by traditional jurists as specific anal intercourse (*liwāṭ*) between men. Classical jurists agree that the *Qur'ān* deals explicitly with sodomy that is illegal, whereas for revisionist scholars, *liwāṭ* has never been defined as a "sexual orientation" as equivalent to the present-day understanding of homosexuality. Hence there is no clear explanation of whether the cause of the offence is truly interpreted as what constitutes the offence of the Lut people. This makes the correlation between *liwāṭ* in *amal qawm Lut (behaviour of the people of Lut)* and homosexuality as it is applied in many modern findings indefensible. For example, a homosexual relationship does not necessarily constitute a penetration. Moreover, *liwāṭ* as anal intercourse has also not been clearly consulted in a heterosexual relationship where some traditional jurists would consider this practice legal (Habib 2010, p. xvii). Additionally, the fact that homosexuality and lesbianism as sexual orientations or identities have never been explicitly condemned in the *Qur'ān* (Rehman and Polymenopoulou 2013; Ozsoy 2021) opens the path to questioning the validity of the classical interpretation that sets punishment for a same-sex sexual act (*hadd*) as an illicit sexual act by traditionalist scholars such as Imam Abu Hanifah. Apart from this reason, the *Hanafi* law does not associate anal sex with *zina* (adultery) and therefore, anal sex in a legal heterosexual marriage does not cause any legal consequence because it is practised within the realm of a legitimate relationship (Kugle 2010a, pp. 189–91). From this *Hanafi* decree, revisionist scholars further believe that a solution for a *halal* (legal) same-sex relationship must be initiated through a marriage, too, just like in a cis-hetero relationship.

This paper also disagrees with the centrist/moderate approach advocated by scholars such as Ramadan (2009). While they manage to distinguish between predilection (desire)

and practice (*liwāṭ*), ultimately such an approach does not support "divine justice" (Esack 1997; Dabashi 2008) nor provide a solid solution for homosexual Muslims. Asking LGBTQ+ Muslims to withhold their sexual desires and refrain from making them public is against the nature of the way these people were created, and it is unfair because while cis-hetero people can resort to marriage, LGBTQ+ people are asked to be patient for rewards in the hereafter. On a social justice level, this opinion is prejudicial, and it is considered not only homophobic but also heterosexist as it against the "human dignity in Islam" (Kamali 2010).

If we further consider all aspects discussed throughout this paper, it might open the possibility of same-sex marriage, at least from a theoretical standpoint of *Hanafi* legal textbooks. As such, there is certainly an urgent need of an *Ijtihād* (legal reasoning) to clarify the legal complexities around the above-mentioned issues. Through the lenses of reformist and progressive Muslims, Lut's story demonstrates that the *Qur'ān* does not dismiss homosexuality in the context of contemporary same-sex relationships but condemns forced sodomy. Hendricks (2010), Kugle (2010b), Siraj (2016) and Ali (2016) read the story through an integrative lens and re-elucidate it by re-examining the language, socio-cultural and historical context in which the text was revealed. Their central analysis of the Lut story is that Lut's people are punished for the accumulation of several sins, the greatest being the rejection of the monotheistic religion and the sexual violence inflicted on other men (Kugle 2010b; Hamzic 2015; Siraj 2016).

*3.2. Conservative and Centrist Muslim Scholars Argue Based on Cis-Hetero Morality Standards*

I have written this paper from the essentialist argument supported by the *Qur'ān* and the *Hadith* (Habib 2008; Kugle and Hunt 2012) to confront the conservatives (Kotb 2004; Al-Qaradawi 2001), who suggest that homosexuality is perverted, illogical, abnormal and immoral behaviour and therefore think that Allah considers homosexuality to be a sexual deviation. This view reflects the outdated European medical theories (read: colonial matrix of power) that define homosexuality as "unnatural", "perversion" and "unhealthy" as opposed to anything "normal", "natural" and "healthy" in heterosexuality (Zollner 2010). While conservative scholars emphasise that heterosexuality is the only natural norm (Al-Qaradawi 2004, pp. 78–80), they have not yet addressed the issue from either medical and psychological (essentialist) or social constructivist reasoning. By default, this opinion is also against the divine law that is always in harmony with the law of nature (Ali 2016) because same-sex attraction can also be natural and essentialist (Kugle 2010b; Ali 2016; Alipour 2017). Al-Qaradawi (2004, pp. 78–80) further called homosexuality "a crime against the rights of females", and unanimously traditionalist scholars agree that "all humans are naturally heterosexual" (Habib 2010, p. 299). This opinion also denies the fact that there are men who are not attracted to women as explained in the *Qur'ān* (24:31). Similarly, this opinion refutes the Islamic decree that a marriage can also be *haram* (prohibited) for an individual when they fear or are certain that they can be unjust to their spouses, i.e., will not be able to fulfil the sexual needs of the spouse.

This paper further disagrees with the stance of Al-Qaradawi (2001) and Kotb (2004), who deny the innate existence of diversity in gender and sexual orientation as illustrated in the *Qur'ān* (42:50; 24:31; 24:60). By their logic, Allah created all humans as heterosexual and therefore only heterosexual marriage is good for everybody. Again, Al-Qaradawi (2001), Bouhdiba (2004) and Kotb (2004) justify their standpoint with an exclusive combination of superficial religious and general cis-heteronormative arguments without addressing the constructivism or essentialism arguments. They further argue against homosexuality as moral and societal problems (e.g., hindering human reproduction) based on direct interpretation of the Lut verse in the *Qur'ān*, concluding that this is disagreeing with humans' fundamental nature (read: cis-heteronormativity). This conservative view has become prevalent as Muslim community leaders preach that homosexuality is an abnormality and a sickness, therefore leaving Muslim LGBTQ+ open to discrimination. While Al-Qaradawi and his allies use a general cis-hetero "morality" argument, in contrast, Kugle and Hunt (2012), Eidhamar (2014), Siraj (2016) and Ali (2016) base their arguments strictly on *Qur'ānic*

premises. They underline the conservative circumvention that seems to ignore the importance of several essential themes in the *Qur'ān*, such as gender equity, diversity in humanity, social justice, the Prophetic example and a forgiving and merciful God.

### 3.3. Lut Story Needs to Be Read as Part of a Cluster, Not in Separation; Thus What Is Condemned in the Qur'ān Is Male Coercive Rape and Domination

This paper concurs with Manji (2005) that the conservative Islam of today tends to follow literalism in reading the *Qur'ān* and this approach has become mainstream due to the prevalence of the ideology of *Wahhabism*. To avoid bias, this paper agrees with the principle that every text in the *Qur'ān* should be interpreted in relation to the *Qur'ān* in its entirety (Jamal 2001; Izutsu 2002; Siraj 2016). In response to the conservative general argument, progressive and reformist scholars (Esack 1997; Jamal 2001; Habib 2008, 2010; Kugle 2010b; Naraghi 2015; Siraj 2016; Ali 2016; Alipour 2017) provide a comprehensive understanding of homosexuality while dismantling and deconstructing the heteronormative portrayal of homosexuality. They further argue that current, predominantly religious, legislation on same-sex acts is based on centuries of archaic heteronormative *Qur'ānic* interpretations and *Shari'a* discourses. As Hendricks (2010, p. 32) concluded:

> Muslims who limit themselves to one interpretation or oppose different interpretations of the *Qur'ān* inhibit the potential of the *Qur'ān* to promote social and spiritual growth. *Qur'ān* 39:55 makes it clear that Muslims are instructed to extract, out of the many possible interpretations, the interpretation that achieves the greatest good.

This paper also disagrees with the conservative accusation presented by Vaid (2017) that claimed that revisionist and progressive Muslims' *Qur'ānic* interpretation demands that the text be reinterpreted based on sexual modernity. This paper affirms that revisionists see the *Qur'ān* as divine speech but because the *Qur'ān* uses a type of language that sometimes can be ambiguous and open to interpretation, it has been continuously reinterpreted, allowing the real message to only gradually reveal itself (Jamal 2001; Izutsu 2002; Siraj 2016). While the tradition of interpretation in Islam is strict, the history of *Qur'ānic* interpretation is also connected to power relations (i.e., patriarchy) and the political interests (e.g., cis-heteronormativity) of the cohorts. In addition, the *Qur'ān* was revealed through its interactive explanatory process, where verses are explained through verses, and therefore the verses need to be observed within a cluster rather than detached from their correlations (Jamal 2001; Siraj 2016). As such, an isolated reading of the Lut story from the conservative point of view has led to a superficial and indiscreet understanding of the current homosexuality issue. For example, by looking at the *Qur'ānic* interpretation of the Lut story as a whole, through *Al-Ankaboot* (29:28–30), the words *innakum la ta'toonal faahishata maa sabaqakum bihaa min ahadin minal 'aalameen* (indeed, you commit such immorality as no one has preceded you with from among the worlds) are mentioned in the verse along with other great sins (including "evil deeds in public assemblies" i.e., *wa tatoona fi nadikum almunkar*, and highway robbery).

### 3.4. The People of Lut Were Mostly Heterosexuals but They Perpetuated Male Violence

The phrase "besides women" (*dūni alnnisa*) that "the Lord has created for you as mates" emphasises that the men of Lut were mainly heterosexuals, often married to women (26:166).[3] Ibn Mansūr's (a traditional jurist) exegesis of the *Qur'ānic* Lot narrative claims that God was chastising the people of Lut in Q 26:165–166 and 7:81 for abandoning their "wives" and instead turning to "males" with coercive lust. They acted impolitely as they approached men with the aggression of rape, deeply rooted in coercion, and these behaviours have nothing to do with contemporary homosexual relationships. On this basis, Kugle (2010a) and Jahangir and Abdullatif (2018) conclude that the verses condemn Lut's people for wanting to forcefully anally penetrate/rape men (the visiting angels) as stated in verses 7:81; 26:165; 27:55; and 29:29. It is important to note that in 27:55, the words *min duna al-nisa'i* (instead of women) are mentioned before *Innakum ta'tuna al-rijala shahwatan* (verily,

you approach men lustfully), indicating that the people of Lut approached men with a lustful intention and not with *mawaddah* (love/compassion). The *fahisha's* explanation of the phrase "*lata'tuna al-rijala shahwatan*" (*Qur'ān* 7:81) is interpreted by al-Tabari (d. 923) as "Lut's people approached onto men in their anus lustfully". Similarly, in verses (29:28 and 7:80), the abomination that "none in the world ever committed before the Lut people" refers to heinous crimes such as sexual assaults along with "evil deeds in public assemblies" and highway robberies. (11:77–78, 11:80, 15:70, 26:169, 54:37).

### 3.5. Verses Al-'Ankabūt (29:28) and Al-'A'rāf (7:80) as Referential Evidence of Male Violence

Verses Al-'Ankabūt (29:28) and Al-'A'rāf (7:80) further show that the abomination "none in the world ever committed before the Lut people" could be used as historical evidence to support the revisionist standpoint. These verses provide a plausible chronological evidence of homosexologics in a historical context by explaining that the condemnation in the *Qur'ān* is for male-on-male coercive rape and domination rather than the condemnation of a mere same-sex relationship that is based on love and compassion. Historically, same-sex unions occurred across the globe long before Prophet Lut's time (c. 1861 or 1761–1687 BCE), as they were thoroughly recorded in ancient Sumeria (c. 4100–1750 BCE), Mesopotamia (c. 3500–530 BCE) and Egypt (c. 3100–48 BCE). In Ancient Egypt, homoeroticism was recorded as early as 2800 BCE, where homosexual relations were common and often extended to a form of marriage (Parkinson 1995). The best-known case of a possible homoerotic relationship is shown in the tomb inscription of pharaoh Niuserre's chief manicurists, Nyankh-khnum and Khnum-hotep. Archaeologists found several paintings depicting both men embracing each other and touching each other nose-on-nose, representing a kiss. (see Figure 1).

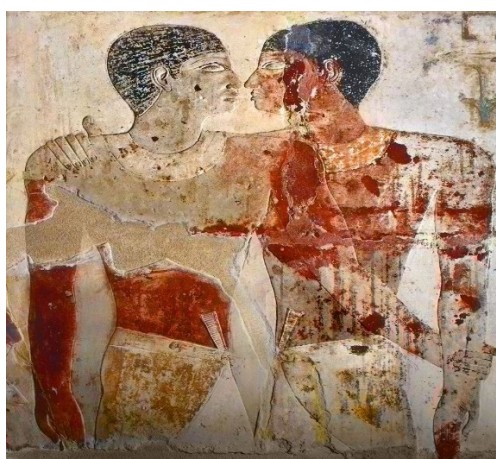

**Figure 1.** Dating from the Fifth Dynasty, the Nyankh-khnum and Khnum-hotep's epigraph reads: 'Joined in life and joined in death'. The presence of their wives and off-spring in certain panels among the glyphs has led some Egyptologists to assume they were siblings. However, repeated depictions of the two men embracing have convinced many Egyptologists that their intimate relationship might have been accepted by their respective wives. One thing is certain: the tomb depicts two men in a manner usually reserved for husband and wife, whilst there were no romantic depictions of their wives with them on the walls of the tomb itself. This argument is strengthened by the position of Khnum-hotep in many of these scenes, placing him in a position normally reserved for women, such as one scene in which he is shown smelling a lotus. Source: Public domain image via the University of Liverpool, https://www.liverpool.ac.uk/archaeology-classics-and-egyptology/blog/2021/sexuality-in-the-past/ (accessed on 27 July 2022). Further reading: Reeder (2000), "Same-sex desire, conjugal constructs, and the tomb of Niankhkhnum and Khnumhotep".

In the famous epic hero Sumerian/Mesopotamian poem "Gilgamesh", which first appeared during the Old Babylonian period circa 1000 BCE, the hero was portrayed in a

homoerotic relationship with one of his best mates, Enkidu. Likewise, in the Almanac of Incantations of Assyrian-Mesopotamia, the text contained prayers for divine blessings for same-sex unions on the same basis as love between a man and a woman (Greenberg 1990, p. 126). Mesopotamia's Code of Hammurabi (written around 1770 BC) mentions Salzikrum women, who are thought to be one of history's earliest mentioned lesbians. Furthermore, a particular Middle Assyrian code from Assur (dating from 1075 BC) condemns homosexual rape or forced sex. For example, table A paragraph 20 deals with a situation where a man (the perpetrator) who has forced sex upon a local resident or business partner should be then punished by castration, indicating the first recorded same-sex rape punishment (Pritchard 2016, p. 181). By default, this code of law runs in parallel with what is condemned in the *Qur'ān*, which is male coercive rape and domination. Following this timeline, Prophet Lut lived around the same time, where homoeroticism itself was not condemned, nor looked upon as immoral or disordered for decades, as long as "false rumours or forced sex was not involved" (Nissinen 2004, pp. 25–27). However, on the level of social constructivism, this is still not considered similar to the contemporary understanding of homosexuality as these laws indicate that it was still shameful for a male to take the submissive woman's role in same-sex intercourse.

*3.6. The Term Fahisha in the Lut Story Does Not Refer to Affectionate Same-Sex Relationships*

In Al-'Ankabūt (29:28–30), the term *fahisha* in the Lut story does not refer to a desire for affectionate love found in today's understanding of consensual same-sex relationships but purely to inhospitality and coercion towards other men. In its comprehensive semantic meaning, *fahisha* is a *Qur'ānic* term for strongly condemned immoral behaviour but it is not necessarily sexual (e.g., *Qur'ān* 3:135, 7:28 and 17:32). As discussed earlier, the *liwāṭ* discourse in the *Qur'ān* excludes the presence of any form of attraction or emotions, pointing exclusively to the act of forced sexual penetration performed with a same-sex partner (Habib 2008; Kugle 2010a). It is not about homosexuality, but rather the Lut people are criticised for using forcible sex as a weapon. As such, the *Qur'ān* (Hud 11:77–79 and Al-Hijr 15:68–72)) narrates how Lut's people intended to rape Lut's guests as the evidence that Lut's people were sexually abusing unwilling men.

So when we read these verses contextualised as a thematic cluster, we gain an understanding of what was actually condemned in the *Qur'ān* through the destruction of Lut's people: (1) sexual violence towards men and highway robberies (7:80; 29:28–30; 26:165); (2) attempts to collectively commit sexual assault on his guests (11:77–80; 15:67–70; 29:33; 54:37); (3) inhospitality to foreigners and guests (26:169; 29:29; 11:77–78; 54:37; 15:68–70); (4) evil deeds in public assemblies (29:29); (5) mocking and denying the [prophetic] warnings (54:33; 29:29); and (6) the rejection of Lut's prophet-hood (7:82; 50:13). The upshot is homosexual rape does not make consensual homosexual conduct wrong any more than heterosexual rape makes heterosexual consensual sex wrong. The rigidity in understanding homosexuality in Islam happens when the traditional scholars construe the Lut story via hermeneutical reading that sexual orientation (same-sex attraction) and the act of *liwāṭ* (anal penetration) are one and the same. Establishing a punishment for male–male intercourse to heterosexual adultery whilst it is not mentioned in the *Qur'ān*, which is already a problematic issue, further raise a question on their legality, making them untenable.

## 4. No Authentic *Hadith* Claims That People Who Engage in Same-Sex Relationships Should Be Killed

As argued above, the *Qur'ān* barely has anything to say about the subject of homosexuality that is relevant to the present time. Unlike the Bible, the *Qur'ān* does not decree earthly punishment for homosexuality. Rather, the historical Islamic basis for justifying the execution of homosexuals is from the *ahadith* (plural of *hadith*). Nonetheless, the authenticity of these *ahadith* has been called into question, especially around the particular text that mention if two men are found committing the act of Lut's people, then both the insertive and receptive partners should be killed (Abu Da'ud 1986). Noteworthy, none of the *ahadith* used

to condemn homosexuals are taken into account by Imam Bukhari and Imam Muslim, who are considered the two most authentic collectors of *ahadith* (Hendricks 2010; Kugle 2010b; Jahangir and Abdullatif 2018). While progressive/revisionist Muslim scholars have never disregarded the *hadith* as a secondary source of Islamic jurisprudence after the *Qur'ān*, the authentication of *ahadith* scorning homosexuality is considered greatly challenging. Almost unanimously, most of these *ahadith* indicate stark and clear weaknesses in the transmission chain (*isnad*) of the text. As explained by principle Muslim *hadith* experts, such as Malik Ibn Anas (d. 795), Abu Dawood (d. 889), Bukhari (d. 870) and Muslim (d. 875), some of the narrators have been deemed untrustworthy and therefore their *ahadith* unreliable (Kamal 2004). As such, even conservative Islamic bodies, such as JAKIM (*Jabatan Kemajuan Islam Malaysia* (English: Department of Islamic Development Malaysia)) (JAKIM 2015, p. 10), have surprisingly considered the *hadith* that says "Whoever is found conducting himself in the manner of the people of Lut, kill the doer and the receiver" as *da'if* (weak) in status. This view is supported by another conservative scholar Shinqiti (2008), that questioned the authenticity of these texts, specifically of those that have the Prophet prescribe the death penalty for *liwāṭ*. Shinqiti (2008) also concluded that no legal punishment is stated in the *Qur'ān* for homosexuality. With so many inconsistencies surrounding the status of each *hadith* that condemns homosexuality, the best way forward is to look at the historical record and stress that there is no evidence that the Prophet punished anyone for homosexuality (Shinqiti 2008; Hendricks 2010). The death sentence set up for homosexual *hadd* in Islam should then need to be obliterated to reflect the absence of such practice/teaching in the lifetime of the Prophet (peace be on him).

## 5. European Colonialism and the Beginning of Revulsion to Homosexuality

> *"Because he is so tender and pretty, we spent that night together as if we were in paradise"*.
> (Abū Nuwās, Abbasid Poet, d. 814)

From an historical point of view, records show that Muslims for centuries were much more tolerant towards homoeroticism than they are today, as a narrative of sexual diversity can be found in many parts of the pre-colonial Muslim world (Duran 1993; Kugle 2001; Bouhdiba 2004; Kligerman 2007). Societies in pre-colonial Islam recognised both erotic attraction and sexual behaviour between members of the same sex even though this attitude often contradicted Islamic law. By arguing that homosexuality is a corrupt Western behaviour, a conservative Muslim may reject the existence of homoeroticism in Islamic history. This dichotomy of the West versus Islam is often indicated by conservative scholars to justify that Islam stands for morality while the West represents a threat to everything thought to be religiously right and morally appropriate. This social reconstruction has been majorly amalgamated within Orientalist homonationalism (Heimer and Dos Ventos 2020), erasing the social existence of "queer Muslims" by placing queerness outside the boundary of Islam. This enforced homonationalism further creates a dichotomy of the us vs. them binary in a way that the West is often regarded as "progressive" and more "tolerant" of LGBTQ groups than the Muslim world.

Ironically, the term "unnatural sex" in reference to homosexuality was only introduced by European Christians in the late 10th century (Kugle 2010b). The Christian crusaders in the Middle Ages called Muslims "permissive" (too liberal) and "sodomitical" (homosexual) (Kugle 2010b) based on the prevalence of homoeroticism, which, in their view, characterised the behaviour of Muslims during that period (Bearman 2012). For the purpose of presenting evidence to support this argument, this paper provides historical archives that show same-sex attraction existed in Islamic civilisations from the earliest times. However, I reiterate that same-sex male-to-male attraction during this time may or may not quite fit into the present-day model of homosexual relationships. On a normality level, it still does not fit into the current homosexuality context as it existed mainly in the realm of hetero-patriarch normativity and power control (money/power/status) between an adult and a younger subordinate. Perhaps the closest resemblance in today's situation is when an arrangement based on money/power/status is made by the rich, older, influential men seeking youthful

rent boys or keeping young boys for sexual pleasure, which also raises an ethical/morality question among Muslims (see also: Shame and Silence: Bacha Bazi in Afghanistan by Sabet (2020)).

In line with constructivism, I further agree with El-Rouayheb (2005) that the concept of homosexuality in pre-colonial and pre-modern Islam was based on active vs. passive roles and (platonic) passion for human beauty expressed in writing vs. lust pursued and that the Islamic jurists condemned the act of sodomy but tolerated homoerotic sentiments. Previously, the majority of the same-sex relationships were established mainly as a reflection of a hetero relationship as it was prevalent that the brides were younger than the husband. Furthermore, it was during the Abbāsid period (752–1258), known as the Islamic Golden Age,[4] that the army from *Khurāsān* practiced homoeroticism widely under this dynasty (Bearman 2012). Likewise, the well-known book *One Thousand and One Nights*, compiled primarily during the Abbāsid era, together with libertine poems of Abū Nuwās are among manuscripts that glorify same-sex promiscuous obscenity (Bearman 2012). Cross-dressing and gender-variant Muslims also lived freely and thrived as they performed at the Abbāsid court, such as in the case of the *ghulamiyyat*, girls dressed as boys in an erotic fashion (Hamzic 2015). (see Figures 2–4).

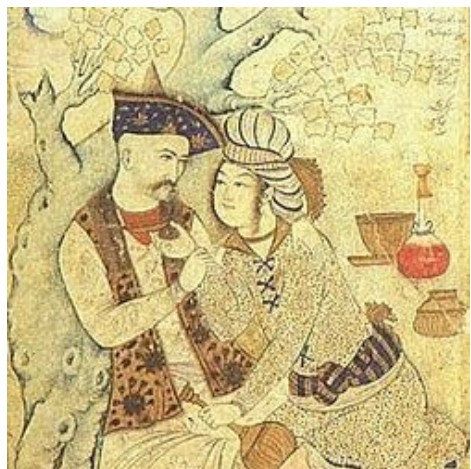

**Figure 2.** The illustration depicts Shah Abbas I of Persia and his page boy interacting and sharing wine, circa 1627. The imagery is soft and intimate as the page holds the wine flask erect towards the Shah's crotch and almost embraces him. Source: Public domain image via https://www. queerarthistory.com/love-between-men/muhammad-qasim-shah-abbas-i-with-a-page/ (accessed on 14 March 2022).

Similarly, in Central and South Asia, homoeroticism was recorded so mundanely in Indo-Islamic society from the 15th century that it was not particularly frowned upon, especially among the Muslim ruling elite (Pallotta-Chiarolli 2020). Babur, the first Mughal Emperor, for example, was known to have had a crush on a boy he saw at a marketplace in Kabul, as explained in his own memoirs, the *Baburnama*. Quite a few Mughal sultans kept young boys in harems for sexual pleasure, and in the public bazaar, young good-looking men could be found dancing everywhere while male prostitutes solicited openly (Kligerman 2007; Daniyal 2016).

Notwithstanding, in Turkey's Ottoman Empire, although there was a radical puritanical orthodoxy movement called the Kadzadeli (1621–1685), which condemned Sufi practices and same-sex attraction, the lack of prosecution and documentation may show that controlling such matters was simply not a priority (Semerdjian 2012). Later, punishment for homosexual intercourse was abolished in 1858 during the *Tanzimat* reforms of the Ottoman Empire by official state scholars from the *Hanafi* School of jurisprudence (Lapidus and Lena 2014; Hamzic 2015). However, a recent study by Ozsoy (2021) argues that such an absence of sodomy/same-sex intercourse as an offense within these criminal codes cannot

be used as a yardstick for decriminalising homosexuality by the Ottomans. Ozsoy (2021) argues that in this new framework of decriminalisation, same-sex intimacy was merely confined to the private sphere and any public reference to it was diminished. Nonetheless, it is largely accepted that it was the 1858 Penal Code that led to the decriminalisation of homosexuality in the Ottoman empire, regardless of whether it was developed through Western legal history or the French Penal Code.

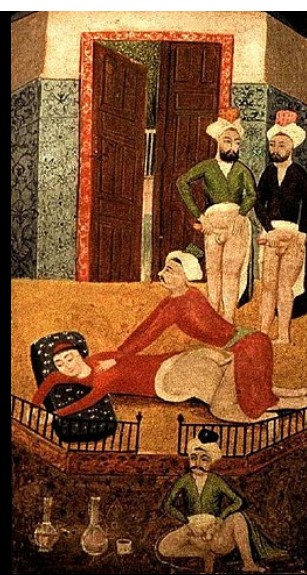

**Figure 3.** The Ottoman Turkish illustration depicting a young man used for group sex (from Sawaqub al-Manaquib); 19th century. Source: Murray (2000, p. 135), Homosexualities, University of Chicago Press, ISBN 978-0-226-55194-4.

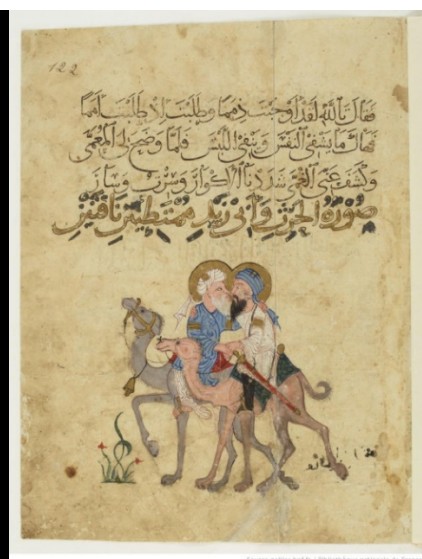

**Figure 4.** Illustration from the 13th century manuscript Arabe 3929; an assembly of stories in a poetic form, known as the Maqamat by Al-Hariri (1054–1122) of Basra (southern Iraq), produced under the late Abbasid caliphate. The manuscript features 99 surviving miniatures highly esteemed, in modern times, for their authentic depiction of the 13th-century Muslim life. In this image, Abu Zayd and al-Harith are riding their camels while holding each other by the shoulder and hand, face to face or, perhaps more accurately, mouth. There appears to be a heart-shaped figure around their heads. Societies in Islam have long recognised both erotic attraction and sexual behaviour between members of the same sex. Picture courtesey of the Bibliothèque nationale de France. See also 'Orality, Writing and the Image in the Maqamat: Arabic Illustrated Books in Context' (George 2012, p. 31).

In Iran, the country that prescribes the death penalty for homosexuality today, literature that contains stories of attraction between men, such as Sadi's classic *Gulistan*, were popular until the early 19th century (Daniyal 2016). Homoeroticism was considered part of regular culture, with some customary Iranian poetry and artefacts containing many explicit and implicit homosexual references. Likewise, Andalusian poetry and traditional Moorish culture have a few references to homoerotic culture (Kligerman 2007), indicating that homoeroticism was not just purely metaphoric in the literature but very much alive in medieval and pre-colonial Islam.

## 6. Homosexuality in Post-Colonial Islam

These above-mentioned homoerotic examples were recorded in Islamic history centuries before homosexuality became accepted in the West, as in the present day. Apart from Islamic radicalisation through its agency, such as *Wahhabism*, colonialism has also played a pivotal role in the homophobic attitude in the Muslim world today. In 1860, Section 377 of the Indian Penal Code was promulgated by the British Raj, making "carnal intercourse against the order of nature" (i.e., homosexuality) an offence in India (Sanders 2009; Daniyal 2016). After that time, Muslim homoerotic culture in India and many other Muslim countries slowly faded as the British Empire grew more powerful in the 19th century (Kugle 2001). This law has become the basis of similar deeply homophobic statutes entrenched in 42 other former British colonies, including Pakistan, Bangladesh, Myanmar, Singapore, Uganda, Malaysia and Brunei. This "unnatural offense" brings a punishment of up to 10 years in prison, with some states, such as Saudi Arabia (read: *Wahhabism*), delivering capital punishment for homosexuality. In Iran, it was only after the Islamic Revolution that same-sex acts became illegal through codification of a particular interpretation of Islamic law. Such a colonial matrix of power (Mignolo 2007), which sought to reconstruct a Muslim/colonised subject by imposing Western sexual norms, ignored local subaltern ideas about culture, moral sensibilities, society and law. The rise of Western colonialism in the Muslim world is correlated with the increasing stigma against homosexuality today (Duran 1993; Pallotta-Chiarolli 2020).

Besides colonialism, the radicalisation of Islam through *Wahhabism* is another post-colonial polemic in the Muslim world. Progressive/revisionist Muslim scholars argue that Muslims have become more conservative in the last century due to colonisation (Kugle 2001; Ghoshal 2010), and this is also linked to the rise of post-colonial radical movements, such as *Wahhabism*, that claim to be more austere versions of Islam (Manji 2005; El Fadl 2014). The rise of Islamist movements since the 1970s has generated attention to the degree of doctrinal influence mostly on sexual moralities and morphologies through asserting control, for example, on the female body and female sexuality. Although Muslims also became more liberal post colonisation, as they accepted nation states instead of Muslim Empires, accepted parliamentary democracy instead of Caliphates and adopted Western suits and ties and many women rejected the veil, conservatism also started with the rise of Islamist groups in the late 1970s and mostly the 1980s as a response to this Arab/Muslim secularism/socialism experiments in the 1950s and 1960s post colonisation. Moussavi and Crow (2005) explain how the *Wahhabi* radical doctrine sponsored by Saudi Arabia (that gained power with the help from the British) is threatening Islam in that it stunts the intellectual diversity of Islam's rational and spiritual legacies while preferring anti-rationalist intellectual minimalism. Majority Muslim animosity towards homosexual practices in the present day has also been part of the political and cultural legacy of European colonialism. The legacy of the colonial matrix of power does not directly express itself today, but it still exists and reproduces itself in the petri dish of the colonised nation in a form of neo-colonialism. By drawing the genealogy and instruments of post colonialism and the current *Shari'a* law interpretation in the context of homosexuality, this paper shows how instilling fear among Muslims through this "divine" law has produced a powerful meta-narrative that reinforces heteronormativity and its repugnance of sexual and gender diversity. Essentially, Rahman (2014) demonstrates that heteronationalism is legitimised

and thrives in many Muslim governments (directly or indirectly influenced by *Wahhabism*) in order to protect these governments' own legitimacy; unsurprisingly, this stance includes embracing homophobia. Muslim authorities today are indebted to the European colonial legacy that allowed for the preservation of patriarchy and androcentric hetero systems by also criminalising homosexuality. They mostly disagree with the Queer Muslim revisionist view that has become a worldwide dialogue because this progressive view challenges their religious patriarchal heterosexual privilege. Colonialism, together with the radicalisation of Islam sponsored by *Wahhabism*, which promotes scriptural literalism and rigid mindless imitation (*taqleed*), has prevented many present-day social questions, including the rights of LGBTQ+, from being addressed. This paper once again emphasises that the traditional patriarchal domination of gender and masculinity is one prominent historical reason why LGBTQ+ people were and are still marginalised and rejected in Muslim society.

However, while so much historical evidence of homoeroticism is recorded in the pre-colonial Muslim world, this paper restates that one should be cautious not to romanticise the pre-colonial Islamic era as glorifying homosexuality. Some religious figures in the past still viewed homosexuality as a sin and same-sex unions were not possible (Kligerman 2007). In 1807, for example, a bizarre case was recorded against a same-sex couple when two men were pushed off from the top of a minaret at the Umayyad Mosque in Damascus, a decision that was considered an aberration among the religious jurists (El-Rouayheb 2005, p. 151). Nonetheless, while homosexuality still stirred the disapproval of several moralists loyal to the tradition of the cis-heterosexual Muslim interpretation, it cannot be denied that same-sex sexual activity had always been endured as part of the social life of pre-colonial and medieval Muslims, as revealed in historical records. Homosexuality was not considered something unusual, and homophobia was less pervasive than it is today. This is incongruous with the views of most contemporary Muslims, who view homosexuality as a Western hedonistic influence. Muslim conservatives today seem to follow the path of colonialism while experiencing cultural amnesia and misunderstanding their own history. In the next section, this paper presents essentialist and constructivist views to support the increasing number of Islamic scholars examining the current understanding of same-sex relationships.

### 7. Homosexuality as a Social Construct (Constructivism) and Inherently Inborn (Essentialism)

> *"A gay or lesbian Muslim is no less than a heterosexual Muslim, except by the intangible criterion of pious awareness of God (taqwa)."* (Kugle 2010a, p. 13, Introduction)

Even among revisionist scholars, there is a dispute regarding their approach about whether "homosexuality" is a social construct (constructivism) or whether it is an adequate term for an innate expression of human sexuality (essentialism). While Alipour (2017) used theological evidence to support his opinion of the essentialism debate, this paper further strengthens his argument with both essentialism and constructivism. By doing this, this paper also refutes Alipour's (2017) argument that Kugle and Hunt's (2012) and Habib's (2008) essentialist epistemology is inadequate to defend all manifestations of same-sex desires and acts as essential or inborn. Both essentialism and social constructivism in Islam are equally credible in the face of historical and theological evidence in supporting same-sex unions. That is, not only does essentialism provide a plausible explanation, but also it is fully compatible with social constructivism and contemporary anthropological evidence.

#### 7.1. Constructivism

Sociology scholars such as Butler (1990), Bersani (1995) and Berg-Sørensen et al. (2010) argued that homosexuality as a recent human sexual practice is a socially constructed phenomenon. Constructivism does not mean accepting that a person's sexual orientation is something they can choose but rather acknowledging that individuals in different cultural or historical settings may establish and tolerate homosexuality. As such, Carlin (1989)

explains that homosexuality in the society developed in parallel to the economic and social foundations after the late 19th century prior to industrial capitalism. I, however, disagree with the claim that homosexuality as a sexual orientation is solely a social construct and that prior to a certain time period people had no knowledge of the concept of homosexuality. I agree with Safron et al. (2007) that homosexuality as a social construct can be construed as a thesis about language epistemology. As such, "homosexual" as a term was introduced in the Western world in the nineteenth century and that word did not exist until the nineteenth century. However, homosexuality in any ontological practise is not necessarily historically culture bound and/or time bound or does not exist only within certain cultures and within certain time periods. Therefore, I agree with Kligerman (2007) that while homoeroticism existed in the past, the society during that time rejected same-sex unions based on traditional understanding to maintain the existing society and the importance of marriage and family in reproduction. Homosexuality then (not producing children) was (and is still) seen as a breach of the "religion of nature". Following the constructivist argument that society has developed, there are a few things that were considered permissible in the *Qur'ān* in the past, but they are no longer considered acceptable in the present day, such as slavery, underage marriage and sexual relationships between an authority and their young subordinate. For this reason, this paper agrees with Carlin's (1989) Marxist position in arguing that the contemporary understanding of homosexuality has a material basis in the urbanisation of human life, allowing people to organise their life around sexual desires and who they choose to be with regardless of their sexual orientation. Some individuals and movements would further connect those desires with their personality and identity. Before the late 19th century, the family was the primary institution by which sexual cis-hetero conformity was imposed. The industrial capitalism of the early 19th century brought multifarious changes, including the polarisation of gender roles for women and men in terms of individuality and personal life—which opened a new era in attitudes to sexuality (Carlin 1989).

*7.2. Essentialism*

Notwithstanding, this paper also agrees with the essentialist epistemology of Muslim revisionist scholars such as Kugle and Hunt (2012) and Habib (2008) that being homosexual has never been a choice; it is an orientation that one can be born with and/or discover over time and precisely therefore part of the Creator's intention. Essentialism is the view that homosexuality is an essential feature of human beings and that it could be found, in principle at least, in any culture and in any time. Kamali (2009) has asserted that both *fiqh* (Islamic jurisprudence) and science confirm that sexual orientation is largely inherent. Whereas science and history have recognised that sexual orientation is not an option but rather can also be an inborn aspect of human sexuality and different sexual orientations have existed since time immemorial (Soble and Power 2008). Moreover, pre-modern Muslim scholars also understood that same-sex attraction could be a "natural" desire. As such, Al Nawawi (d. 1278) stated that a "male youth is like a woman as his beauty is similar to a woman's beauty and that he is desired as she is desired" (cited in El-Rouayheb 2005, p. 114). Ramadan (2009), a contemporary Muslim scholar, further states that if this attraction does not represent some moral deviance or manifest some abomination, it is still tolerable because it is "natural". All the same, Islamic reformist El Fadl (2002) and Kugle (2010a) maintain that *Qur'ānic* verses such as 49:13, 42:8, 42:49:50, 11:118–119 and 30:22 show that sexual diversity among human beings is part of the world that God created, and the *Qur'ān* does not condemn men with such desires. The revelation "those who are not procreative" may refer to lesbians and gays (42:50); those who are "neither male nor female, or "mix of males and females" or are "in between" could refer to twins but could also refer to transgender and intersex people (42:50); and "men who have no sexual desire with women" may refer to eunuchs, impotent men, non-heterosexuals, gays, transgenders and asexual people (24:31); while "the women who are not reproducing and do not wish for intercourse" in 24:60 may relate to lesbian women. Moreover, *Qur'ān* (36:36) also recognises the likelihood of God to create people in pairs of possible partners who we know about

(male and female) and who we do not know about. This could explain the diversity of partnerships among human beings and that heteronormative marriage is not the only viable option. Not only that, the terms for "mates" or "partners" in the *Qur'ān* are more than once mentioned in a non-gender-specific way (Hendricks and Krondorfer 2011). Kugle (2010a); Kugle and Hunt (2012) and Habib (2008) argue that since the *Qur'ān* accepts diversity as part of God's creative will and if same-sex desires and acts are part of that diversity, they are indeed accepted by the *Qur'ān*.

This essentialism argument, however, does not sit well with the centrist and conservative views. Although moderate/centrist Muslim scholars may have accepted different gender identities and expressions, the issue of committing homosexual behaviour still challenges them. Centrist and traditional scholars, such as Vaid (2017), claim that innate homosexuality is equivalent to the impulse to lie, steal or cheat by saying that these are also "natural" impulses but similar to the homosexual impulse, acting upon them is a sin and therefore prohibited. They further argue that if homosexuality is going to be allowed in Islam, it will open a floodgate for other sins, such as alcoholism, to be legalised based on the natural urge. It seems unscientific to compare an elementary need for intimacy to an addiction or "corrupt" behaviours. Homosexual desire, just like heterosexual desire, when it is mutually managed, does not cause harm or damage to anyone as opposed to those "dishonest" behaviours. Indeed, when an individual is deprived of sexual relationship, as a need, it could cause distress and be destructive to the individual. In the present day, to argue for same-sex unification to be suppressed for the public good because it posits a threat to a cis-hetero normative relationship is irrelevant and contrasting with the constructivism argument throughout this paper. Essentially, this will leave LGBTQ+ people without a reasonable alternative, which violates the Islamic ethos of human dignity and justice (Jahangir and Abdullatif 2018) and leaves them to suffer from the mental health consequences of such suppression. Prescribing celibacy to persons with homosexual desires does not provide them a legitimate way to satisfy their human sexual, emotional and physical desires. After all, permanent celibacy has never been part of Islamic teaching. Further, forcing Muslim homosexuals to marry the opposite gender is haram (unlawful) as it causes suffering and unfairness to both spouses.

## 8. Homosexuality Today as Both Essentialism and Constructivism: A Same-Sex Union Possibility in Islam

Although the modern term "homosexuality" only emerged as a concept in the late 19th century due to the new social realities of industrial capitalism (Carlin 1989), homoeroticism has been long recorded in the history of humankind as same-sex inclination is naturally innate (essentialism). The difference lies in the way in which the new social reality of homosexuality has gradually become an alternative relationship to heteronormativity as society developed (constructivism), as we can observe in the present day. In the past, homosexuality was considered unsupportable as a sexual act or as same-sex union and it was regarded as sinful or criminal. The traditional understanding of a relationship has always been informed by religious power domination represented by the hetero-male gender, the rich and the educated that inserted their ubiquitous heteronormativity (Bourdieu 1996; May et al. 2014). In addition, Tolino (2014) argues that in medieval and pre-modern times, the main criterion of a marriage was not sexual orientation (binarism hetero/homosexuality) but the type of role each person was supposed to play in a sexual relation, i.e., that the man was the active partner, while the passive partner was a woman, either a wife or a concubine, or even a young boy or a slave. Together with the "literal" interpretation of homo-related texts in the *Qur'ān* and the Bible, such a structure has prevented the homosexual minority from living openly in accordance with their innate identity, especially in the past century (Ahmed 2006).

For these above-mentioned reasons, this paper argues that not only can essentialism be perceived as a credible theory, but that essentialism and constructivism can co-exist when it comes to supporting the present same-sex unification in Islam. Indeed, same-sex unions

should not be regarded as sinful or criminal anymore in the present day given the historical evidence and the invariance that homosexuality as constructivist has developed, as well as the current understanding from revisionist Muslim scholars on Prophet Lut's story. Considering that sexual desire is naturally created by Allah (essentialism), a resolution to homosexuality must be found through marriage in order to make same-sex relationships legal. Given the fact that the centrist/conservative view has only been made based on a superficial and exclusive cis-heteronormative discourse, and in their inability to rebut the essentialism and constructivism argument, there are good reasons to follow in the footsteps of revisionist Muslim scholars. In addition, if conservative scholars can agree to comparing homosexuality to cis-hetero adultery, then a definition of marriage in the present day must be inclusive of same-sex marriage so that homosexual people can avoid being sinners. Comparing same-sex marriage to cis-hetero marriage in Islam, it allows justice, equality, prevention of harm and removal of hardship (Kamali 2005, p. 166), as the *Qur'ān* also provides liberation, rejects prejudices and delivers strength to the oppressed and marginalised (Karmi 1996). Noteworthy, Kamali's (2005) view is supported by Islamic scholars such as Ibn Taymiyyah (1998), who held that when a decree causes more harm than good (e.g., no marriage for homosexuals leading to celibacy or being promiscuous), such a ruling must be reconsidered. Hence, this paper argues that the traditional understanding of marriage in Islam as solely a contract between a man and a woman to make vaginal intercourse legal in the eyes of God should be given an extended inclusive understanding. A marriage must include same-sex relationships in the present day to reflect the universal concept of marriage in Islam. As such, the *Nikah* (marriage solemnisation) as described in the *Qur'ān* serves as a protection (2:187), a companionship (16:72) based on mutual consent (4:19), love and care (30:21–22) so that the couple may find tranquillity between themselves (30:21). As Kamali (2005, p. 162) explains, given that consent alone creates obligatory rights and responsibilities, a form of Muslim same-sex marriage is also foreseeable. After all, ordaining for *liwāṭ* the same punishment as for heterosexual adultery came from an interpretation of earlier traditionalist scholars who lived in a time when the current form of homosexual love was not accepted and family structures were based on marriages between men and women, with procreation as an important goal.

## 9. Conclusions

This paper has presented a re-examination of the Lut story as it is deemed important to address the issue of contemporary homosexuality and provide a space for LGBTQ+ Muslims to practise their religion peacefully. This paper has also addressed relevant issues related to the pre-colonial and post-colonial legacies of current Muslim LGBTQ+ sexualities and how European colonialism forced its way into the Muslim world. This paper then used the arguments of Jamal (2001), Manji (2005) and Siraj (2016) to call for a reform-minded position that gives hope for non-hetero Muslims to live by their identity while still staying within the Islamic religion. This call is urgent, especially when many LGBTQ+ Muslims are struggling with their spirituality while dealing with their homosexual orientation or gender identities, including intense guilt due to the inability to stop committing "unforgivable sins". This paper also commits to Manji's (2005) call to unshackle the colonialist mentality, which allows Saudi Arabia's *Wahhabism* to conquer the minds of Muslims, by critiquing *Wahhabism*'s scriptural literalism and rigid mindless imitation (*taqleed*). This call is also important to re-open the door of *Ijtihād* (reasoning) in Islamic jurisprudence that has been replaced by *taqleed*, which controls religious tolerance. In the absence of *Ijtihād*, many present-day social problems, including homosexuality, cannot be addressed and Muslims have been blaming the West for their misfortunes.

Notwithstanding, conservative Muslim leaders have failed to address the issue of the human need for intimacy and affection for non-hetero Muslims when they are left with difficult choices that cause more problems, such as alcohol abuse, internalised homophobia and suicidality. Furthermore, if the conservatives can agree on the assumption that God is just by nature, they should be able to agree with progressive and reformist Muslim scholars'

elucidation of the Lut story. Especially, following the constructivist argument that society has developed, there are a few things considered permissible in the *Qur'ān* in the past that are no longer considered acceptable, such as slavery and underage marriage. To date, this research has not encountered any explanation by conservatives to counter the essentialism (and constructivism) argument on homosexuality that the reformist/progressive Muslims have presented.

In conclusion, this paper enlightens the readers to another approach of seeing a possible same-sex union in Islam that has been broadly discussed in Jahangir and Abdullatif (2016, 2018) and widely supported by other Muslim reformist scholars, such Hashim Kamali, Farid Esack and Abd al-Razzāq Kāshānī. Their arguments are based on the principle of "human dignity in Islam and affection" (Kamali 1999, 2010), constructed on "Allah's universal love and grace to his creatures" (Kāshānī 1991) and the "divine justice" of liberation theology (Dabashi 2008; Esack 1997) as well as "Islamic diversity and tolerance" (El Fadl 2014; Stepan and Taylor 2014). Above all, they follow a specific "paradigm shift in traditional method of *Ijtihād*" (Alipour and Hasani 2011) of examining Islamic theological-juridical methodologies and concepts in their effort to prevent harm and facilitate a legal outlet for sexual expression as stated in *Qur'ān* verse 4:28. Ultimately, this paper enlightens both cis-hetero and sexually diverse Muslims through a pertinent Islamic sexual discourse in the hope of empowering them as well as allowing them to reconcile around their religious, legal, political and social stances.

**Funding:** This research received no external funding.

**Conflicts of Interest:** The author declares no conflict of interest.

## Notes

[1]  Hudud (sg. hadd) literally means limitations and is a legal technical term for offences with fixed, mandatory punishments that are based on the *Qur'ān* and *ahadith* (sg *hadith*). This includes, for example, theft, banditry, unlawful sexual intercourse, unfounded accusation of unlawful sexual intercourse and drinking alcohol. However, similar to adultery, homosexuality as a *hadd* crime is hard to prove, given the need to provide four untainted witnesses, and therefore it is a private sin (Bearman 2012).

[2]  Some upper-class men that acted as receptive partners were viewed as suffering from the *ubnah* disease. See the "Treatise on Hidden Illness" note by Ar-Razi, a prominent medevial Muslim pyshcian (Rosenthal 1978). Ar-Razi noted that such men might be cured through enemas and massages by good-looking maids. Ar-Razi describes *ubnah* as being derived from weak male sperm that makes the male child effeminate. Nonethelss, Ar-Razi demonstrates that homosexuality was viewed as a natural, genetic phenomenon.

[3]  The *Qur'ānic* phrase that describes men who approach other men lustfully "instead of women" (*min dun al-nisa*) actually means "besides the women", indicating that these men might behave heterosexually with their wives. See http://corpus.quran.com/qurandictionary.jsp?q=dwn#(27:55:6) (accessed on 1 May 2022) (cited from Siraj 2018).

[4]  During the Abbāsid period, the Muslim world became an intellectual centre for science, astronomy, alchemy, mathematics, medicine, philosophy and education, in which the House of Wisdom (Grand Library of Baghdad) was a place where both Muslim and non-Muslim scholars sought to translate and gather all the world's knowledge into Arabic. During this period, the Muslim world was a cauldron of cultures that collected, blended and advanced the knowledge gained from the Roman, Chinese, Indian, Persian, Egyptian, North African, Ancient Greek and Medieval Greek civilisations (Gregorian 2003).

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
