# Peer review of "Reconsidering Homosexual Unification in Islam: A Revisionist Analysis of Post-Colonialism, Constructivism and Essentialism"

_religions, doi:10.3390/rel13080702_

Round 1

Reviewer 1 Report

Please see the attached document for a detailed review of the paper. 

In addition to those points there are a couple more:

1) 728 – 730: awkward wording. What you are saying it marriage should not be applicable between a man and a woman, but what you want to say is that it should not be only applicable between a man and a woman.

2) 780 -781: facilitating a legal outlet 780 for sexual expression as stated in Q_u_r_’ān_ _verse 4:28 this has to be referenced with Jahangir and Abdullatif (2018), as the specific understanding and argument comes from there

PS: I am writing these two here, as the system does not allow to upload an updated version of the document. 

Author Response

Thank you for your review and please find the reply in attachment.

Reviewer 2 Report

I read the article with interest. It is a challenging and thoughtful text that will provoke quite some debate. Not all readers will agree with you but your way of discussing and analyzing is acadamically sound. 

Author Response

Thank you very much for your review.

However, since your comments are mainly on syntax and grammars, I have addressed them directly in the paper. 

Reviewer 3 Report

Please see in the attachment.

Author Response

(The authors gave the same response as above.)

Round 2

Reviewer 1 Report

The draft is massively improved and strengthened. It is a much better and stronger article now. I recommend publication after spell check and grammar usage. 

This manuscript is a resubmission of an earlier submission. The following is a list of the peer review reports and author responses from that submission.

Round 1

Reviewer 1 Report

  1. The author tried to cover a wide range of topics, and as a result, s/he was not able to elaborate his/her arguments and clarify his/her claims adequately.
  2. The author needs to cite his/her sources properly.  For example, in many cases, it is not clear that the author refers exactly to what part(s) of the sources.  In many cases, there is no page number!
  3.  The author heavily relied on the authority of other scholars without adequately discussing their arguments.  
  4. The author's general approach to the topic seems more polemic than scholarly.   The paper could be a good polemic piece to challenge some of the misconceptions among conservative Muslims, but not necessarily as a solid piece of scholarly work.
  5. The author raised an expectation that s/he is going to offer a new approach to the issue on the basis of Jahangir and Abdullatif, however, there is no detailed and extensive discussion of those scholars' views, and it is not clear how the author achieved that goal.   
  6. In many cases, the author is not careful and precise in his/her claims.  For example, the author says: "The majority of contemporary Muslims today follow the radical interpretation of the Quran found in Wahhabism and in certain Sunni and Shia sects."  The claim that "the majority " of contemporary Muslims follow "the radical interpretation of the Qur'an" is not supported, and more importantly, it is not clear what the author means by "radical interpretation". Or  the author writes, "Islam has recorded a long history of normalising homoeroticism and sexual behaviour."  However, the idea that "Islam" has been normalizing homoeroticism is very much questionable.  There seems to be confusion between "Islamic" and "Islamicate".
  7. The author claims that "the term liwat (sodomy) is mentioned in the Quran", but this claim is not accurate.
  8. The author's discussion on essentialism vs. constructivism is rather hasty, and it is not clear in what sense s/he synthesizes these two conflicting views.  
  9. It is not quite clear why the author refers to evidence of erotic love for children and youngsters in the context of homosexuality.  At the very least, this may lead to confusion, as if the author conflating homosexuality and pedophilia.  
  10. The author writes that in Islam, "the master was permitted to make equal use of his male slaves as he did of the female (Boronha, 2014)" To the best of my knowledge, Islamic jurisprudence (Sunni and Shii) does not approve of free men having sex with their male slaves.  So I am not sure what the evidence for this claim is.  

Reviewer 2 Report

The article does not have a clear argument, a clear research question, and a clear structure. You put together different aspects related to homoeroticism and homosexuality in Islam and in Islamicate societies, both in the pre-modern and in the modern period, but the golden thread cannot be followed. I also wonder what is the original research in this paper.  

Also, the state of the art is not up to date. Following fundamental works for the article have been ignored, for example Khaled El-Rouayheb 2005;  Amr A. Shalakany 2000; Camilla Adang, 2003; Sara Omar 2012; Serena Tolino: 2014; 2020; 2021. 

The way the author tries to reconcile essentialist and constructivists arguments seems not substantiated to me and unconsistent.

There are also a number of inaccuracies:

p. 2: it is not true that most medieval  scholars except Hanafis believed the punishment for liwat should be equal to the hadd for zina. There are at least two opinions between shafi'i and hanbalis too).

p. 2: the author says: "the term liwat in the Quran..:" : the term liwat is never mentioned in the Quran. Moreover, Peri Bearman is not the author of the entry on liwat in the EI2, only the editor of the EI2. 

p. 3: it is mentined al-Bukhari 2000. What exactly?

p. 3: "being gay and Muslim.... have existed since time immemorial and same-sex attraction is naturally inherent". This is highly essentialist. How do you reconcile this with the constructivist argument? 

p. 3: studies on homosexuality between animals are mentioned, but no reference is given.

p. 4: "before the late 19th century, the family was the primary instition by which sexual cis hetero conformity was imposed". The nuclear family is a modern product! There is a lot of historical research on that.

p. 5 and  in other parts of the article: a number of Quranic verses are mentioned. It would be necessary to add them in the footnotes. 

p. 5, before paragrah 3: there is a "they" which does not make sense.

Inconsistencies: sometimes you write Lot, sometimes Luth. Why Luth with a final h? Similary, you use sometimes LGBTQ sometimes LGBT+.

at p. 9 you mention ahadith to condemn homosexuals. This is also essentialist and difficult to reconcile with the constructivist debate.